# A Study of the Molecular Regulatory Network of *VcTCP18* during Blueberry Bud Dormancy

**DOI:** 10.3390/plants12142595

**Published:** 2023-07-09

**Authors:** Ruixue Li, Rui Ma, Yuling Zheng, Qi Zhao, Yu Zong, Youyin Zhu, Wenrong Chen, Yongqiang Li, Weidong Guo

**Affiliations:** 1College of Life Sciences, Zhejiang Normal University, Jinhua 321004, China; lrx2497809053@163.com (R.L.); marui7202@163.com (R.M.); zyl732078048@foxmail.com (Y.Z.); zhaoqi17353517996@163.com (Q.Z.); yzong@zjnu.cn (Y.Z.); cwr@zjnu.cn (W.C.); 2Zhejiang Provincial Key Laboratory of Biotechnology on Specialty Economic Plants, Zhejiang Normal University, Jinhua 321004, China; 3School of Agricultural, Jinhua Polytechinc, Jinhua 321007, China; zhuyouyin@jhc.edu.cn

**Keywords:** blueberry, *VcTCP18*, *BRC1*, yeast one-hybrid, upstream regulatory gene, bud dormancy

## Abstract

*BRANCHED1* (*BRC1*) is a crucial member of the *TEOSINTE BRANCHED1/CYCLOIDEA/PCF* (TCP) gene family and is well known for playing a central role in shoot branching by controlling buds’ paradormancy. However, the expression characteristics and molecular regulatory mechanism of *BRC1* during blueberry bud dormancy are unclear. To shed light on these topics, shoots of three blueberry cultivars with different chilling requirements (CRs) were decapitated in summer to induce paradormancy release and subjected to different levels of chilling in winter to induce endodormancy release. The results showed that the high-CR cultivar ‘Chandler’ had the strongest apical dominance among the three cultivars; additionally, the expression of *VcTCP18*, which is homologous to *BRC1*, was the highest under both the decapitation treatment and low-temperature treatment. The ‘Emerald’ cultivar, with a low CR, demonstrated the opposite trend. These findings suggest that *VcTCP18* plays a negative regulatory role in bud break and that there may be a correlation between the CR and tree shape. Through yeast 1-hybrid (Y1H) assays, we finally screened 21 upstream regulatory genes, including eight transcription factors: zinc-finger homeodomain protein 1/4/5/9, MYB4, AP2-like ethylene-responsive transcription factor AINTEGUMENTA (ANT), ASIL2-like, and bHLH035. It was found that these upstream regulatory genes positively or negatively regulated the expression of *VcTCP18* based on the transcriptome expression profile. In summary, this study enriched our understanding of the regulatory network of *BRCl* during bud dormancy and provided new insights into the function of *BRC1*.

## 1. Introduction

Bud dormancy is a vital adaptive mechanism that allows plants to survive harsh environments. It can be categorized into three types: paradormancy, endodormancy, and ecodormancy [1]. During paradormancy, the breaking of axillary buds is repressed by apical dominance, which determines shoot branching and influences the architecture of plants, ultimately affecting the number of flowers and fruits. Therefore, the yield of trees is closely related to branching [2,3,4,5]. For perennial plants, endodormancy is a crucial progression over winter that determines the quality of bud break, flowering, and fruiting [6]. Although these types of dormancy are often viewed as occurring separately, any given bud may be controlled simultaneously by any or all of the signals regulating these aspects of dormancy [7].

Blueberries (*Vaccinium* spp.) are highly valued for their fruit, which are rich in anthocyanins [8]. Field observations have shown that different blueberry cultivars exhibit significant variation in their chilling requirements (CRs) and shoot branching. Cultivars with high chilling requirements, such as ‘Brigite’, ‘Chandler’, and ‘Duke’, typically exhibit strong apical dominance, characterized by limited branching and upright shoot growth. Conversely, cultivars with low chilling requirements, such as ‘Sharpblue’, ‘Emerald’, and ‘Jewel’, exhibit weak apical dominance, a high capacity for axillary bud break, abundant branching, and spreading shoots. This pattern has also been observed in other species, such as sweet cherry (*Prunus avium* L.) and Chinese cherry (*Cerasus pseudocerasus* L.). Shoot branching, which is related to the paradormancy of the axillary buds, and chilling requirements, which are related to the endodormancy of flower buds, are the two critical agronomic traits of fruit trees. These traits play crucial roles in determining plants’ structure and yield, respectively, in the coming year. However, is there any correlation between these seemingly independent traits?

Bud arrest and outgrowth are complex processes that are influenced by various factors, including environmental and developmental cues [9]. The regulation of bud break and branching is mainly controlled by a key gene, *TB1*, which was first discovered in maize (*Zea mays*) [10]. The *TB1* ortholog is also found in rice, known as *OsTB1* or *FINECULM1*, and in *Arabidopsis*, pea, and tomato, known as *BRANCHED1* (*BRC1*). *BRC1* encodes a TCP transcription factor that acts locally in buds and serves as a crucial hub of different signals that control bud break [11,12]. Research has shown that *BRC1* is regulated by strigolactones (SLs) and sugar, and it can promote the expression of the *PIN3* gene, which plays a dominant role in maintaining apical dominance and the plant’s architecture [13,14]. In poplar, studies have shown that *BRC1* is regulated by dormancy-associated MADS-Box (*DAM*) and regulates the expression of *NCED3*, a key gene involved in abscisic acid (ABA) synthesis. This suggests that *BRC1* inhibits bud break and plays an important role in buds’ endodormancy [15]. In addition, short-day (SD)-induced *BRC1*-like factors inhibit shoots’ apical bud growth by binding to the FT-like protein FT2 and antagonizing its growth-promoting activity [16]. In potato, *BRC1a* and *BRC1b* promote axillary bud dormancy [17,18]. Despite the different signals and hormones involved in buds’ endo- and paradormancy, the core regulatory network appears to be the same, with *BRC1* acting as a central hub for controlling the buds’ outgrowth.

In our previous research [19], we identified the homologous gene *VcTCP18* of *AtBRC1* in blueberry through bioinformatics and analyzed its function through genetic transformation. It was found that *VcTCP18* not only inhibits the occurrence of lateral bud break in *Arabidopsis* but also inhibits seed germination. Since the expression of *BRC1* is influenced by both endogenous (hormonal and nutritional) and exogenous (light) inputs, it serves as the hub of bud break; however, the differences in the *BRC1* expression levels and patterns among cultivars with different CRs are still unclear. Moreover, the upstream regulatory genes of *VcTCP18* and its molecular network, which regulates the dormancy of blueberry buds, is yet to be fully understood. Therefore, in this study, we aimed to explore the expression patterns of *VcTCP18* in different cultivars during the release of paradormancy and endodormancy, analyze the possible *cis*-acting elements in the gene promoter, observe the promoter activity of *VcTCP18* by its transient expression in tobacco, and screen the upstream regulatory genes through yeast one-hybrid screening. This helped to enrich the regulatory network of *VcTCP18* and provide insights for plant genetics and breeding.

## 2. Results

### 2.1. Expression Patterns of VcTCP18 in the Process of Para- and Endodormancy in Different Blueberry Cultivars

On 17 June 2019, a decapitation treatment was applied to the shoots of three different blueberry cultivars. Following a period of 20 days, the extent of axillary bud breakage was observed and quantified. The findings revealed that ‘Emerald’, an evergreen Southern Highbush Blueberry (SHB) cultivar characterized by a low CR, exhibited significant germination of a large number of axillary buds. Notably, ‘Emerald’ demonstrated the highest bud germination percentage compared with the other two cultivars, as visually represented in Figure 1A.

On the basis of field observations over multiple years, it was found that the shoot tips of ‘Emerald’ were often damaged by borers during the summer season. This damage caused the axillary buds to break, even though the terminal buds appeared to be normal. These observations suggested that the apical dominance of ‘Emerald’ is weak, and the apical bud has a weak ability to inhibit the lower axillary bud. ‘O′Neal’ is a deciduous SHB cultivar with a moderate CR that exhibited a stronger apical dominance compared with ‘Emerald’, with only two or three axillary buds breaking after decapitation. In contrast, the Northern Highbush Blueberry (NHB) cultivar ‘Chandler’ exhibited a high CR and typically showed breakage of one or even no axillary buds after decapitation. This observation suggests that ‘Chandler’ displays a stronger apical dominance compared to ‘Emerald’ and ‘O′Neal’.

The expression of *VcTCP18* in the buds of the three blueberry cultivars was analyzed using qRT-PCR before and after decapitation. Figure 1B shows that ‘Emerald’ had the lowest expression of *VcTCP18*, and there was no significant difference in its expression before and after decapitation. In ‘Chandler’, the expression level of *VcTCP18* was observed to be higher in dormant axillary buds before decapitation and increased sharply 2 days after decapitation. In contrast, ‘O′Neal’ exhibited a higher expression level of *VcTCP18* compared with ‘Emerald’ but lower than that of ‘Chandler’ before and after decapitation. Overall, the expression levels of *VcTCP18* varied among the three cultivars; however, it was observed that there was a negative correlation between the gene expression levels and axillary bud breakage.

As illustrated in Figure 2A, during the process of endodormancy release, the expression levels of *VcTCP18* were downregulated in all three cultivars with increased chilling, indicating that the expression of *VcTCP18* was inhibited by low temperatures. Among the cultivars, ‘Chandler’ exhibited the highest expression level of *VcTCP18*, followed by ‘O′Neal’; ‘Emerald’ had the lowest expression level. The bud breakage rates of the three cultivars are shown in Figure 2B. On 17 November, the bud breakage rates of all cultivars were relatively low, not exceeding 50%, indicating that the flower buds were in the endodormant state. On 8 December, the bud break rate of ‘Emerald’ reached 67%, indicating that endodormancy had been released. In contrast, the bud breakage rate of ‘O′Neal’ was 16%, and ‘Chandler’ showed the lowest bud break rate of only 1.2%. These results indicated a negative correlation between the expression level of *VcTCP* and the breaking of flower buds in the blueberry cultivars. Furthermore, the CRs of the cultivars, ranked from high to low, were ‘Chandler’, ‘O′Neal’, and ‘Emerald’. This finding suggests that the CRs of blueberry cultivars are associated with the expression levels of *VcTCP18*.

### 2.2. Promoter Activity Analysis of the VcTCP18 Gene

Cis-acting elements play a significant role in the regulation of gene transcription initiation. These elements are crucial because they contain specific DNA binding sites and other regulatory motifs that interact with transcription factors. We selected promoters within the 3000 bp region upstream of 5′UTR for analysis using PlantCare (http://bioinformatics.psb.ugent.be/webtools/plantcare/html (accessed on 13 September 2020)). The *cis*-acting elements of *VcTCP18* are shown in Appendix A, including a variety of elements such as photo-responsive elements, drought-induced MYB-binding sites, active elements involved in the salicylic acid reaction, and common cis-acting elements in promoter and enhancer regions, among others.

In order to analyze the activity of the *VcTCP18* promoter, the 2860 bp sequence upstream from the start translation codon ATG and two different 5′-end deletion mutants (Figure 3A) were cloned into the pMD 19-T vector and transformed into *E. coli* DH5α. The recombinant plasmid vectors were successfully constructed and transformed into *Agrobacterium*, which was verified by bacterial liquid PCR (Figure 3B).

An *Agrobacterium* strain containing Pro/P1/P2-*VcTCP18*-PGreen II 0800LUC recombinant plasmids was injected into the back of tobacco leaves; empty PGreen II 0800 LUC was used as a negative control. Fluorescence analysis demonstrated that the *VcTCP18* promoter exhibited activity, and the expression activity of Pro*VcTCP18*, P1*VcTCP18*, and P2*VcTCP18* showed a gradual decrease (Figure 4A).

Previous studies conducted in our laboratory revealed that *VcTCP18* exhibits high expression levels during the deep endodormancy stage of blueberries. However, when subjected to low temperatures, the expression levels of *VcTCP18* decrease rapidly [19]. In order to investigate the regulatory effect of low temperature on the expression of *VcTCP18*, we performed a luciferase reporter assay using promoters containing different missing fragments under 25/4 °C treatments. The results of the fluorescence analysis showed that the *LUC* gene retained its expression activity but with a lower fluorescence intensity than the control group at normal temperatures. Further measurements of the *LUC*/*REN* ratio (Figure 4B) indicated that the promoter activity of pro/p1*VcTCP18* was significantly inhibited at low temperatures, while the expression activity of p2*VcTCP18* showed no significant change compared with the control group at normal temperatures. This suggests that the T1 (−2860 bp to −2310 bp) and T2 (−2310 bp to −1595 bp) segments of the *VcTCP18* promoter are crucial for its response to low temperatures.

### 2.3. Screening of A Yeast One-Hybrid Library

To further investigate the regulatory genes upstream of *VcTCP18*, the bait vectors *VcTCP18*-T1-pAbAi and *VcTCP18*-T2-pAbAi were constructed. To prevent the self-activation of the yeast cells’ own endogenous transcription factors with cis-acting elements of the target gene promoters, it was crucial to perform background level detection prior to screening the cDNA library. The results showed that neither bait vector showed any growth at an AbA concentration of 50 ng/mL or higher, indicating the absence of self-activation (Appendix A).

After the cDNA library was screened, positive plasmids from the growing yeast strains were selected and further analyzed. On the basis of the sequencing results, we excluded clones with undetectable sequences or unknown functions. Our analysis revealed that four proteins were able to bind to the *cis*-acting elements of the *VcTCP18*-T1 promoter (Table 1), namely, Class III chitinase, ASIL2-like, glyceraldehyde-3-phosphate dehydrogenase GAPCP1, and selenocysteine methyltransferase. Furthermore, for the *VcTCP18*-T2 promoter, 16 proteins were found to interact with its cis-acting elements, as shown in Table 2. Among these, eight were transcription factors, i.e., zinc-finger homeodomain proteins 1/4/5/9, dehydrin 1, MYB4, AP2-like ethylene-responsive transcription factor AINTEGUMENTA (ANT), and BHLH03.

### 2.4. Verification of the Positive Clones by Rotation

The 21 positive plasmids obtained from the Y1H screening were co-transformed into yeast with *VcTCP18*-T1-AbAi and *VcTCP18*-T2-AbAi and grown on SD/-Leu and SD/-Leu/50 ng/mL AbA media for validation by rotation. The resulting colonies were further validated through dilution assays, where the newly grown monoclones were diluted 1, 10, 100, and 1000 times.

As illustrated in Figure 5, the positive control (p53-pGADT7 co-transformed with p53-pAbAi) grew successfully on both media. Conversely, the negative control (pGADT7-AD co-transformed with *VcTCP18*-T1-pAbAi and *VcTCP18*-T2-pAbAi) grew on SD/-Leu after four dilution gradients but failed to grow on SD/-Leu/50 ng/mL AbA, as expected, indicating the validity of the control system. 

Almost all of the 21 candidate-positive clones were able to grow on both the SD/-Leu and SD/-Leu/50 ng/mL AbA media after dilution, indicating their ability to interact with the *cis*-elements of the *VcTCP18* promoter.

### 2.5. Analysis of the Expression of the Screened Genes

The transcriptome data were utilized to validate the expression profile of the candidate genes, as depicted in Figure 6. The dates 19 November, 8 December, and 29 December represent the states of endodormancy, endodormancy release, and ecodormancy, respectively. The time points of 6, 12, 18, and 24 h correspond to the process of ecodormancy release. Cluster analysis revealed that the expression patterns could be divided into two distinct groups. The first group displayed a similar expression pattern to *VcTCP18* and primarily consisted of transcription factors (TFs) such as zinc-finger homeodomain protein 1/4/5/9, MYB4, AP2-like ANT, and ASIL2-like. These genes exhibited high expression levels during the endodormancy stage (19 November), followed by downregulation during endodormancy release (8 December). Subsequently, their expression levels gradually increased during the process of ecodormancy release (warm treatment for 6, 12, 18, and 24 h). The expression pattern of the second category was characterized by low expression levels during endodormancy, followed by a gradual increase with the increase in chilling, reaching the highest expression level and then decreasing during ecodormancy and thereafter. These findings suggested that these genes act as negative regulators of the expression of *VcTCP18*.

## 3. Discussion

### 3.1. Bud Break and BRC1

The process of shoot branching is predominantly determined by the formation of axillary buds, followed by their breaking and growth. The formation of axillary buds is controlled by genetic factors [20], while the breaking and growth of axillary buds are regulated by a complex interplay of factors, which exhibit a high degree of plasticity in response to various cues, such as the availability of sugar, hormone signaling, and changing environmental conditions [21,22]. The breaking of axillary buds ultimately determines the degree of branching and thus impacts the morphology, yield, and productivity of crops and trees [3,4,5].

Apical dominance refers to the suppression of axillary buds’ growth by the apical bud. While auxin has traditionally been viewed as the main inhibitor of axillary buds’ growth, recent research has highlighted sugar as the primary signal in apical dominance. For instance, experiments involving the application of sugars such as sucrose, pectin, and fructose to the stems of roses under light have shown that they promote the growth of lateral buds, with glucose having no significant effect [23]. As illustrated in Figure 7, plant hormones also play crucial roles in the regulation of bud breaking, with auxin and strigolactone (SLs) inhibiting bud growth and cytokinin (CTK) promoting bud growth. The channelization model suggests that auxin is transported to the main stem in a polar manner during the synthesis of young leaves at the shoot tip, ultimately leading to the indirect inhibition of branching.

Multiple pathways converge on the transcription factor *BRC1*, which is expressed in buds and serves as a central regulator, inhibiting bud break and controlling shoot branching in plants [24]. As depicted in Figure 7, *BRC1* exerts its regulatory effects on bud break via two distinct mechanisms. Firstly, BRC1 represses the expression of *PIN3*, resulting in the decreased efficiency of auxin transport and ultimately leading to the accumulation of auxin in the axillary buds, consequently inhibiting bud break. Secondly, BRC1 upregulates the expression of *NCED3*, which is a pivotal gene involved in the synthesis of ABA, thereby increasing the ABA content and leading to the inhibition of bud break. Therefore, *BRC1* plays a critical role in modulating the signaling pathways underlying bud break.

In the present study, we observed that the expression pattern of *VcTCP18* during paradormancy release was similar across different blueberry cultivars, but the expression levels varied significantly. Notably, the upright cultivar ‘Chandler’ exhibited a high expression level of *VcTCP18* after decapitation, resulting in a severe inhibition of lateral bud break. In contrast, there was no significant change in the expression of *VcTCP18* in ‘Emerald’ before and after decapitation, leading to a higher rate of lateral bud breakage. The moderately branched cultivar ‘O′Neal’ displayed intermediate levels of *VcTCP18* expression and bud breakage. These findings are consistent with a previous study on cucumber, which showed that the expression of *CsBRC1* was higher in cultivated cucumber (moderately branched) than in its wild ancestor (highly branched) [13]. The results preliminarily indicated that the ability of axillary buds to break was related to the expression level of *VcTCP18* in blueberry.

Overall, these results indicate that blueberry cultivars differ in their level of apical dominance, which may have important implications for their growth and development. This information is valuable for blueberry growers and breeders when selecting appropriate cultivars for their specific requirements. Further research is necessary to elucidate the underlying mechanisms of apical dominance in blueberry and to determine how these mechanisms can be utilized in cultivation practices.

Compared with paradormancy release, the focus during endodormancy release shifted from axillary buds to flower buds. However, the expression level of *VcTCP18* remained consistent across the different cultivars. Specifically, the expression level of *VcTCP18* was observed to be high during the endodormancy stage and decreased with the increase in chilling. Cultivars with high CRs exhibited a high initial expression level of *VcTCP18*, while those with low CRs had low initial expression levels of *VcTCP18*.

Studies have shown that *BRC1* is positively regulated by *DAM* and is thus involved in responses to low temperatures. During the dormancy cycle, the differential expression of *DAM* genes was observed in the buds of two pear cultivars, ‘Cuiguan’ and ‘Suli’, which have low and high CRs, respectively. The expression level of *DAM* genes was found to be approximately 10 times higher in ‘Suli’ compared with ‘Cuiguan’ [25]. The expression of dormancy genes, such as *DAM* and *BRC1*, was found to be associated with the blueberry cultivars exhibiting different chilling requirements for bud release. This relationship may have been influenced by breeding. SHB were bred by crossing NHB blueberries with wild evergreen blueberries. We speculate that during the hybridization process, the chromosomes underwent recombination, leading to a potential weakening of the regulatory effect on dormancy genes.

According to the results of the experiment, it was found that *VcTCP18* plays a crucial role in the paradormancy and endodormancy of blueberries, and it has a negative impact on bud break. Furthermore, the expression level of *VcTCP18* is closely related to the level of dormancy. Therefore, it can be inferred that there may be a correlation between shoot branching and chilling requirements in perennial woody plants.

### 3.2. BRC1 and Its Upstream Regulatory Genes

The *BRC1* gene can be directly or indirectly influenced by various environmental factors, such as light, temperature, and other factors. For example, SD triggers the downregulation of the expression of FT2, consequently leading to the downregulation of LAP1, a known repressor of BRC1. This downregulation of LAP1 subsequently induces the expression of BRC1. Importantly, BRC1 interacts physically with FT2 and acts as an antagonist, further enhancing the effect of SD and ultimately inhibiting bud growth [16]. A previous study conducted on poplar demonstrated that low temperatures led to a reduction in ABA levels and suppressed the expression of the transcription factor SHORT VEGETATIVE PHASE-LIKE (SVL). Consequently, the expression of the downstream *SVL* gene *TCP18*/*BRC1* was inhibited, ultimately promoting bud break and the transition from dormancy to active growth [15]. In the present study, the activity of pro*VcTCP18* and p1*VcTCP18* was found to be significantly inhibited under low temperatures, while there was no significant difference in the expression activity of p2*VcBRC1* compared with normal temperatures. These findings suggest that the promoter region (−2860 bp to −1595 bp) is crucial for the response of the *VcTCP18* gene to low temperatures. The experimental results indirectly demonstrated that low temperature can suppress the expression of *VcTCP18*, thereby reducing its inhibitory effect on bud break, leading to the release of bud dormancy.

In this study, 21 positive clones were screened using yeast one-hybrid screening, many of which were found to be related to bud dormancy, including *ANT*, four members of the zinc finger homeodomain protein family, abscisic acid stress-induced maturation protein (ASR), and *MYB4*. Among these genes, ANT belongs to the ERF family, which is a large family of transcription factors that feature the conserved AP2/ERF domain and play key roles in various aspects of plants’ growth, development, and stress responses. Transcriptome analysis revealed that the changes in the expression of *ANT* were similar to those of *VcTCP18*.

Furthermore, dual luciferase assays showed that *VcANT* could increase the activity of the promoter of *VcTCP18*, indicating that it could promote the expression of *VcTCP18*. The photoperiod, ethylene, and abscisic acid signal transduction pathways are known to consecutively control bud development by setting, modifying, or terminating these processes [26]. Ethylene followed the photoperiod response and preceded the ABA signal during the induction of bud dormancy. It was speculated that ethylene induces the expression of *ERFs*, thereby regulating *VcTCP18*, increasing the content of ABA, and inducing the buds to enter endodormancy. Recent evidence strongly supports the notion that the accumulation of chilling triggers a significant downregulation of the ethylene content. Additionally, it has been observed that the expression levels of *ERFs* and *BRC1* also decrease under chilling conditions [19,27]. In our study, we observed a similar trend, with the expression of *VcTCP18* decreasing and the ability of flower buds to break being enhanced. This provides further evidence for the internal molecular mechanism of the release of endodormancy, which may be attributed to the downregulation of ethylene and ERFs.

MYB transcription factors are among the largest families of transcription factors in plants and are involved in various abiotic stresses [28,29,30,31]. The overexpression of *MYB4* has been shown to increase the cold and drought resistance of rice and apples [32,33]. Transcriptomic data have revealed that the expression pattern of *MYB4* is similar to that of *VcTCP18*. It was speculated that as the photoperiod changes, the expression of *MYB4* in flower buds increases, which enhances a plant’s cold resistance and helps it to withstand the upcoming low temperatures. According to the transcriptome data and previous studies, it is possible that *MYB4* plays a role in upregulating the expression of *VcTCP18* in the regulation of bud dormancy. *Abscisic* acid *stress ripening* (ASR)-*like protein* is a highly hydrophilic, small-molecule, thermostable protein whose expression is induced by ABA and stress [34]. A study in sugarcane [35] showed that the expression of *ASR* is significantly upregulated in response to the exogenous application of ABA or cold stress, with a rapid decline following the initial upregulation. The expression level of *ASR* was high in blueberry during the early stage of endodormancy (19 November) (Figure 6), whereas the expression level of *ASR* was markedly low at the stage of endodormancy release (8 December), which was consistent with the expression patterns of *VcTCP18*. In conclusion, the photoperiodic signal of short days triggers the induction of dormancy in woody plants. As a response to the upcoming cold winter, a cascade of genes that resist cold stress are expressed, promoting the expression of *BRC1*, increasing the content of ABA, and ultimately inhibiting bud break.

In contrast to genes that are downregulated during the release of endodormancy, certain genes identified through yeast one-hybrid screening displayed an expression pattern that increased with the accumulation of chilling and decreased with the accumulation of warm temperatures. This expression pattern was completely opposite to that of *VcTCP18*, suggesting that these genes may inhibit the expression of *VcTCP18*. In turn, the inhibition of the expression of *VcTCP18* promoted the release of bud dormancy. Among these genes, *dehydrin* is very interesting, as it is usually produced under stress conditions such as low temperatures, high salt, and drought, which cause cell dehydration. In our research, two *DHN* genes were identified, indicating that they have a strong ability to bind to the promoter of *VcTCP18*. *EjDHN5* was found to be induced by low temperatures in loquat [36]. The overexpression of *EjDHN5* can significantly improve the cold resistance of tobacco [37]. At the early stage of endodormancy (19 November) in blueberries, there was no increase in chilling and, as a result, the expression level of dehydrin was relatively low. With the increase in chilling, the buds experienced low temperatures, which promoted the expression of *dehydrin*, thereby inhibiting the expression of *VcTCP18*. This observation may explain why increased chilling is necessary for the release of bud endodormancy.

## 4. Materials and Methods

### 4.1. Plant Materials

The plant materials for this study were obtained from the blueberry orchard (29°1′39.05″ N, 119°44′18.17″ E) of the Zhejiang Provincial Key Laboratory of Biotechnology on Specialty Economic Plants in China. Three blueberry cultivars, namely, ‘O′Neal’ and ‘Emerald’ from the SHB group, and ‘Chandler’ from the NHB group, were selected for the study. The regions suitable for growing the NHB are in the northern parts of the United States, and its chilling requirements are high (600–1200 h of chilling between 0 and 7 °C) [38]. In order to expand the range of adaptation of the NHB, an important breeding goal is to reduce chilling requirements. The SHB are hybrids between highbush and evergreen blueberries, which has resulted in progeny with lower chilling requirements than the highbush parent (100 to 600 h). ‘O′Neal’ is a variety that was developed in the 1980s and 1990s and has a chilling requirement of 400 to 500 h [39]. ‘Emerald’ was selected around the year 2000 and has a chilling requirement of 100 to 400 h [40].

On 17 June 2019, when the buds entered paradormancy, the shoots were decapitated, and 20 axillary buds were collected before and 2 days after decapitation. These buds were immediately frozen in liquid nitrogen before being stored at −80 °C as samples from the paradormancy release period.

In 2020, samples from the endodormancy release period were collected by pruning shoots with flower buds from the three blueberry cultivars on 17 November and 8 December. These dates were chosen to represent different stages of endodormancy release of the blueberry cultivars. Shoots collected from multiple plants were pooled as a single sample, and samples collected from different locations in the field were used to create three biological replicates [41]. The flower buds were frozen immediately in liquid nitrogen and stored at −80 °C.

This study used the plant expression vector pCAMBIA1381z; the yeast bait vector pAbAi; the prey AD vector pGADT7; *Nicotiana benthamiana*; 100 μM acetyleugenone; the GUS staining kit (Beijing Kulaibo Technology Co., Ltd., Beijing, China); anhydrous ethanol; the yeast strain Y1HGold; the yeast media minimal SD base, minimal SD agar base, -URA DO Supplement, and -LEU DO Supplement (Beijing Kulaibo Technology Co., Ltd., Beijing, China); YPDA; *BstB* I; salmon sperm DNA; 0.9% NaCl; 1.1 × TE/LiAc; DMSO; Aureobasidin A (AbA); Matchmaker Insert Check PCR Mix 1 (Baori Medicine Technology (Beijing) Co., Ltd., Beijing, China); etc.

### 4.2. qRT-PCR Analysis

Total RNA extraction and 1st-strand cDNA synthesis were performed according to a previous report [42]. The HiScript III RT SuperMix for qPCR (+gDNA wiper) (Nanjing Nuoweizan Biotechnology Co., Ltd., Nanjing, China) was used for the reverse transcription of 1 μg of RNA according to the manufacturer’s instructions. Synthesized first-strand cDNAs were diluted 3-fold for qPCR validation. Specific primers (Appendix A) were designed using Primer-BLAST (https://www.ncbi.nlm.nih.gov/tools/primer-blast/ (accessed on 5 April 2021)), and *VcGAPDH* was used to normalize the amount of cDNA among the samples [43]. qPCRs were performed on an ABI Step One PlusTM RT-PCR system (Applied Biosystems, Waltham, MA, USA). The PCR system, procedures, and data analysis were performed according to [44]. Data analyses were performed using the relative quantitative method (2^−∆∆Ct^). SPSS Statistics 21 software (IBM, Armonk, NY, USA, SPSS Statistics 21) was used for significant difference analyses (one-way ANOVA followed by Duncan’s test), and Prism was used for drawing.

### 4.3. Prediction of the Promoter’s Cis-Acting Element

The 3000 bp upstream of the coding regions of *VcTCP18* gene was extracted using TBtools software [45]. Plantcare (http://bioinformatics.psb.ugent.be/webtools/plantcare/html (accessed on 13 September 2020) was used to search for potential *cis*-acting elements [46].

### 4.4. Promoter Activity Analysis

Total DNA was extracted from blueberry flower buds using the CTAB method [47] and was stored at −20 °C.

The range from −2860 bp to −1 bp was selected as the full-length sequence of the *VcTCP18* promoter. Different promoter lengths were deleted depending on the distribution of the key *cis*-acting elements, and three variants were created: pro*VcTCP18* (−2860 bp to −1 bp), p1*VcTCP18* (−2310 bp to −1 bp), and p2*VcTCP18* (−1595 bp to −1 bp). The primer sequences are shown in Appendix A. Cloning was performed using genomic DNA as a template. The promoters of *VcTCP18* were amplified using PrimeSTAR Max DNA Polymerase and then cloned into the pMD 19-T vector, transferred into *E. coli* DH5α, verified with PCR, and finally transferred into the pCAMBIA2300−3 × flag vector via double digestion and recovered with T4 DNA ligase. The recombinant plasmids pGreenII 0800-LUC-pro*VcTCP18*, pGreenII 0800-LUC-p1*VcTCP18*, and pGreenII 0800-LUC-p2*VcTCP18* were transferred into the *Agrobacterium* strain GV3101; the empty pGreenII 0800-LUC vector was utilized as a negative control. An infection solution of *A. tumefaciens* was injected into the back of tobacco leaves to spread the bacterial liquid fully in the leaves. After injection, the tobacco was cultured in a normal environment for 3 days.

Fluorescence was detected using an in vivo imaging system (Tanon-6600, Shanghai, China). The activity of LUC/REN was detected using the Dual-Luciferase^®^ Reporter Assay System (Promega, Madison, WI, USA). Each experiment was conducted in triplicate, and the results are reported as the mean value ± standard deviation (SD).

### 4.5. Screening of the Y1H Library

Two segments of the promoter sequence of *VcTCP18*, namely, T1 (−2860 bp to −2310 bp) and T2 (−2310 bp to −1595 bp), were cloned into the pAbAi vector. The resulting bait plasmids were named *VcTCP18*-T1-pAbAi and *VcTCP18*-T2-pAbAi, respectively.

For better integration into the Y1HGold yeast strain, the bait plasmids were linearized by the *BstB* I enzyme. The empty pAbAi and p53-AbAi vectors were used as negative and positive controls, respectively. The target bands were recovered and purified using the AxyPerp^TM^ PCR Cleanup Kit (Aisijin Biotechnology (Hangzhou) Co., Ltd., Hangzhou, China) [48].

The Matchmaker Insert Check PCR Mix 1 (Baori Medicine Biotechnology (Beijing) Co., Ltd., Beijing, China) was used to identify the positive bait yeast, and 1% agarose gel electrophoresis was used to detect the PCR amplification products. These primary positive colonies were picked and grown in 600 μL of YPDA liquid medium at 30 °C and 200 rpm for 16–18 h. The supernatant was discarded after centrifugation, and 1 mL of the YPDA liquid medium and 1 mL of 50% glycerol were added. The mixture was stored at −80 °C for later use.

To prevent the endogenous transcription factors in yeast cells from binding to the *cis*-acting element of the promoter, the yeast strains were plated on SD/-URA media containing different concentrations of AbA (0, 50, 100, 200, 400, and 600 ng/mL). After incubation at 30 °C for 3–5 days, the minimum concentration of AbA that could completely inhibit the growth of the yeast strains was determined and used for further library screening.

A plasmid of the cDNA library of ‘O′Neal’ blueberry flower buds was constructed by Shanghai Ouyi Biomedical Technology Co., Ltd. (Shanghai, China) and transformed into the bait yeast. Then, 150 μL of the transformed strain solution was spread on SD/-Leu/50 ng/mL AbA plates and incubated for 3–5 days at 30 °C until single clones with sizes of 1–2 mm were obtained.

### 4.6. Identification of Positive Clones and Extraction of the Prey Plasmids

The single clones that grew well and were of a large size on the SD/-Leu/50 ng/mL AbA medium were carefully picked and mixed with 5 μL of ddH_2_O. Colony PCR was performed using the 3′ AD and 5′ T7 primers.

The single clones with clear colony PCR bands were selected and transferred into a new SD/-Leu/50 ng/mL AbA medium and cultured for 3–5 days at 30 °C to obtain well-grown colonies. To confirm positive interactions, single colonies were re-streaked twice, and colony PCR was performed on the resulting colonies to gradually eliminate false positives.

The selected positive single clones were expanded in 5 mL of the SD/-Leu/50 ng/mL AbA liquid medium. The yeast plasmids’ DNA was extracted using the Yeast Plasmid Mini Kit (OMEGA, Bridgeport, NJ, USA) and transformed into *E. coli* DH5α, and the successfully transformed strains were stored for verification via rotation and for sequencing.

### 4.7. Positive Clone Rotation Verification

The screened prey plasmids, the empty pGADT7 vector (negative control), and the empty p53-pGADT7 vector (positive control) were transferred into competent yeast cells containing the bait vector. The transformed yeast strains were plated onto SD/-Leu and SD/-Leu/50 ng/mL AbA media and incubated for 3–5 days at 30 °C. The positive clones, which exhibited sizes ranging between 2 and 3 mm, were subsequently suspended in 20 μL of ddH_2_O and diluted 1, 10, 100, and 1000 times to enable further investigation.

Next, 4 μL of the diluted yeast strains was streaked onto both SD/-Leu and SD/-Leu/50 ng/mL AbA media, followed by incubation for 3–5 days at 30 °C. The colony’s growth was observed and recorded for further analysis.

### 4.8. Expression Pattern of the Upstream Regulator Genes in the Transcriptome

The expression patterns of the screened genes were analyzed using the transcriptome data from ‘O′Neal’ blueberry flower buds during endodormancy and ecodormancy release. The genes’ expression values were calculated using a range of zero to one, and the expression heatmap of the screened genes was generated using TBtools software [45].

## 5. Conclusions

Shoot branching and CRs are important agronomic traits in plants, as they are closely associated with bud break, which is, in turn, regulated by *BRC1*, a key factor in the process. In this study, we examined the expression levels of the homologous gene *VcTCP18* of *AtBRC1* in the process of the paradormancy and endodormancy release of buds in three blueberry cultivars that exhibit different levels of branching and CRs. The results further support the notion that *VcTCP18* acts as a negative regulator of bud break, and the expression levels of *VcTCP18* are closely related to the ability of shoots to branch and the CR levels. Additionally, we predicted that there may be a correlation between shoot branching and CRs in woody plants. By analyzing the promoter of the gene, it was found that its activity decreased with low temperatures. Twenty-one regulatory genes upstream of *VcTCP18* were obtained by yeast one-hybrid screening. These regulatory genes coordinately regulate *VcTCP18* in response to environmental factors such as photoperiod and low temperatures to complete the induction and release of bud dormancy. This study greatly broadened our knowledge of the mechanism and regulatory network of *BRC1* in bud dormancy.

## Figures and Tables

**Figure 1 plants-12-02595-f001:**
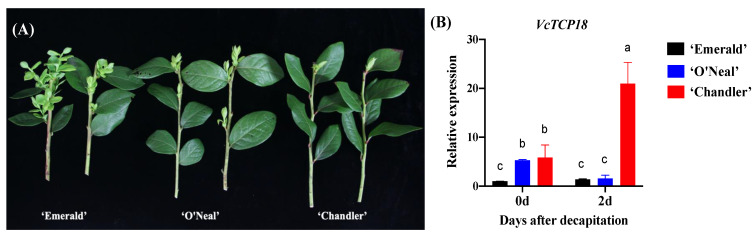
Bud breakage and the expression of *VcTCP18* after decapitation. (**A**) Comparison of axillary bud breakage in the three blueberry cultivars after decapitation. (**B**) Expression levels of *VcTCP18* in the three blueberry cultivars before and after decapitation. Error bars represent ± SD, and a, b, and c indicate significant differences among the means (*p* < 0.05).

**Figure 2 plants-12-02595-f002:**
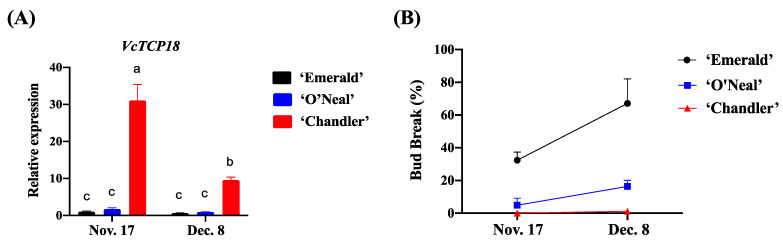
Relative expression levels of *VcTCP18* and the bud breakage rate during the process of endodormancy release. (**A**) Expression levels of *VcTCP18* in the three blueberry cultivars in the process of endodormancy release. Error bars represent ± SD. a, b, and c indicate significant differences at *p* < 0.05. (**B**) Comparison of flower bud breakage among the three blueberry cultivars.

**Figure 3 plants-12-02595-f003:**
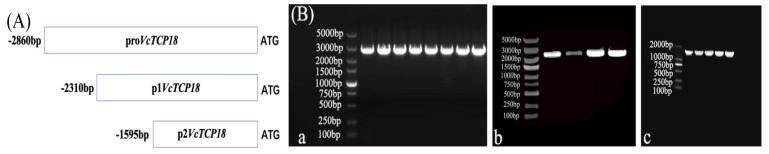
(**A**) Diagram of the 5′−end deletion analysis of the *VcTCP18* promoter. (**B**) PCR validation of *Agrobacterium.* Electrophoretogram of the expression of the *VcTCP18* promoter in the vectors: (**Ba**) pro*VcTCP18*. (**Bb**) p1*VcTCP18*. (**Bc**) P2*VcTCP18*.

**Figure 4 plants-12-02595-f004:**
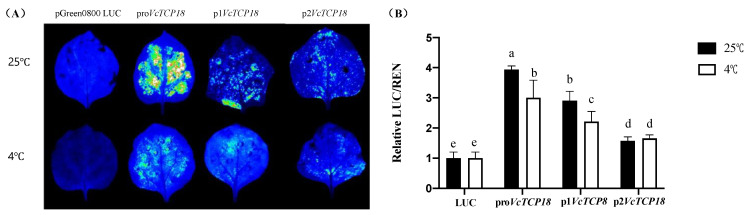
Analysis of the activity of the *VcTCP18* promoter in response to low temperatures. (**A**) Verification of the expression activity of pro/p1/p2-*VcTCP18*-LUC under the 25 °C and 4 ℃ treatments. (**B**) Relative fluorescence expression of pro/p1/p2-*VcTCP18*-LUC under the 25 °C and 4 °C treatments. Error bars represent ± SD. Different letters indicate significant differences at *p* < 0.05. Note: LUC: empty pGreen II 0800 LUC vector.

**Figure 5 plants-12-02595-f005:**
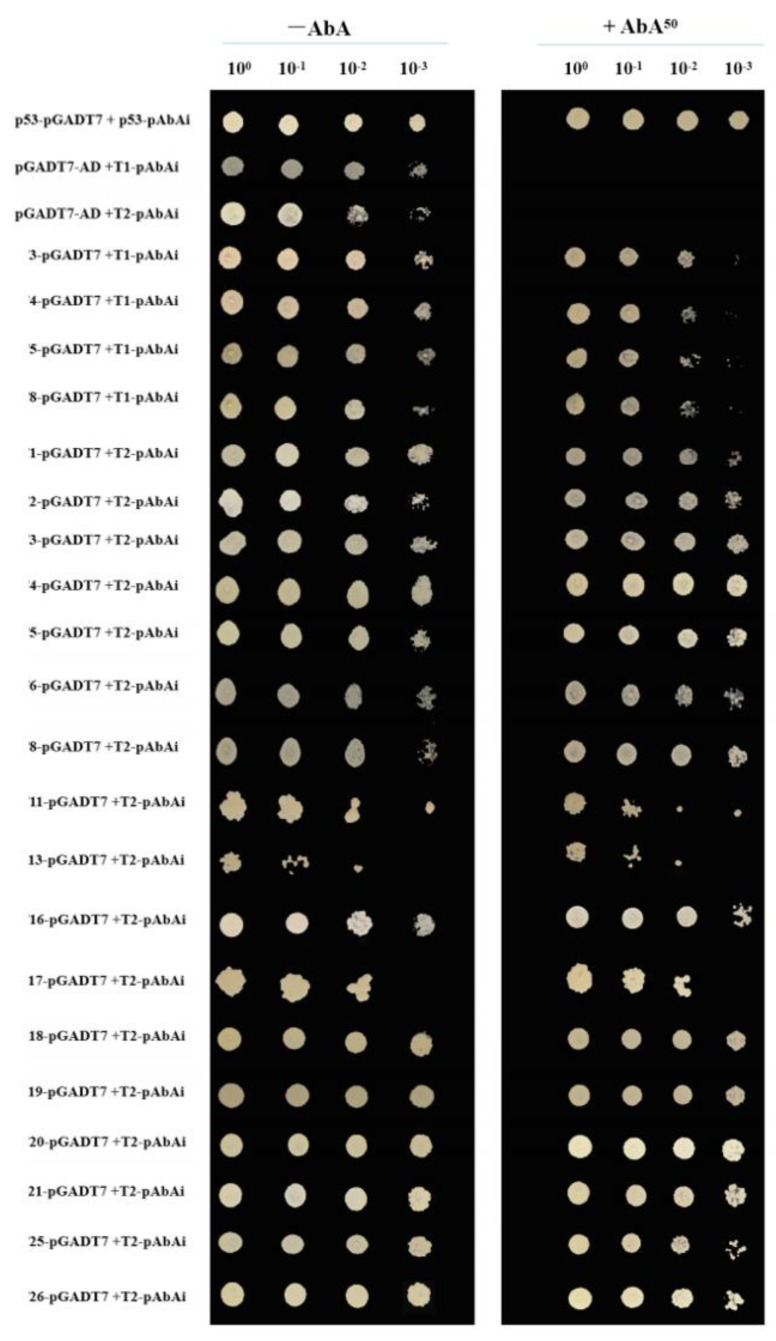
Validation of the positive clones by rotation. The positive transformants were determined by spotting the serial dilutions (1:1, 1:10, 1:100, and 1:1000) of yeast onto SD/-Leu and SD/-Leu/50 ng/mL AbA plates. T1 and T2: truncated *VcTCP18* promoters (located at −2860 bp to −2310 bp and −2310 bp to −1595 bp upstream of *VcTCP18*, respectively). Positive control: p53-pGADT7/p53-pAbAi; negative control: pGADT7-AD/*VcTCP18*-T1-pAbAi and the pGADT7-AD/*VcTCP18*-T2-pAbAi promoter.

**Figure 6 plants-12-02595-f006:**
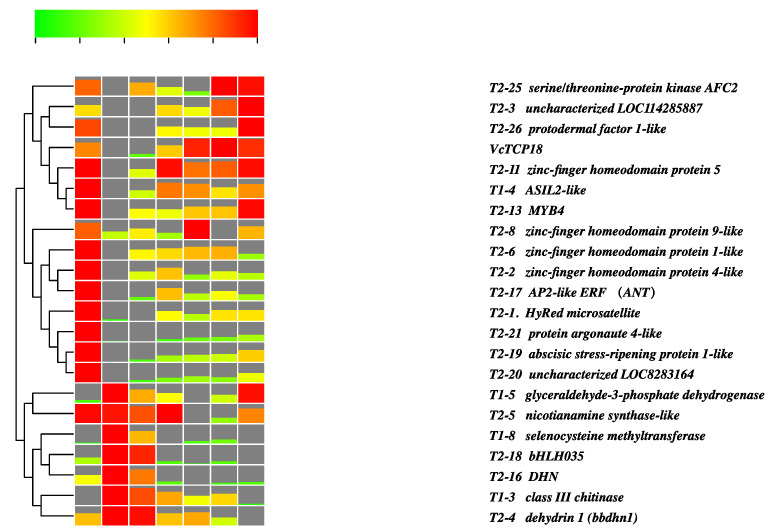
Expression profiles of *VcTCP18* and its potential upstream regulatory genes in the transcriptome of flower buds during dormancy release. On 19 November: flower buds entered endodormancy; on 8 December: flower buds started endodormancy release; on 29 December: the flower buds entered ecodormancy. The ecodormant flower buds were subjected to a warm treatment for 6, 12, 18, and 24 h, and these timepoints represent the release of ecodormancy.

**Figure 7 plants-12-02595-f007:**
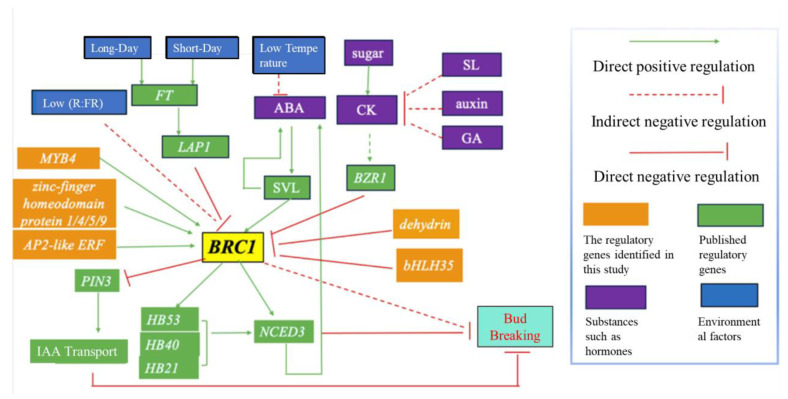
Hypothetical model of *BRC1*’s involvement in bud break. This proposed model represents a simplified review of the multiple pathways by which *VcBRC1* regulates bud break.

**Table 1 plants-12-02595-t001:** Sequencing results of positive clones detected by Y1H[*VcTCP18*-T1-pAbAi] screening.

Serial Number	Gene Name	NCBI Login Number	Identity	E-Value
3	*Class III chitinase*	GU944517.1	84.00%	0.0
4	*Transcription factor ASIL2-like (LOC109148174)*	XM_019296010.1	80.89%	1 × 10^−58^
5	*Glyceraldehyde-3-phosphate dehydrogenase GAPCP1*	XM_028206951.1	81.20%	0.0
8	*Selenocysteine methyltransferase*	DQ480337.1	88.27%	0.0

**Table 2 plants-12-02595-t002:** Sequencing results of positive clones determined by Y1H[*VcTCP18*-T2-pAbAi] screening.

Serial Number	Gene Name	NCBI Login Number	Identity	E-Value
1	*HyRed microsatellite*	KP279095.1	84.83%	2 × 10^−25^
2	*Zinc-finger homeodomain protein 4-like*	XM_016035315.2	76.37%	2 × 10^−88^
3	*Uncharacterized LOC114285887*	XM_028228973.1	82.82%	2 × 10^−88^
4	*Dehydrin 1 (bbdhn1)*	AF030180.1	96.09%	0.0
5	*Nicotianamine synthase-like*	XM_028241468.1	83.10%	1 × 10^−141^
6	*Zinc-finger homeodomain protein 1-like*	XM_018974724.1	85.57%	9 × 10^−48^
8	*Zinc-finger homeodomain protein 9-like*	XM_030657350.1	77.68%	6 × 10^−46^
11	*Zinc-finger homeodomain protein 5*	XM_010062653.2	87.36%	2 × 10^−46^
13	*MYB4*	KT225482.1	99.24%	0.0
16	*DHN*	KF192819.1	99.60%	0.0
17	*AP2-like ethylene-responsive transcription factor (ANT)*	XM_028207531.1	86.88%	0.0
18	*bHLH035*	KU933649.1	95.48%	0.0
19	*Abscisic stress-ripening protein 1-like*	XM_035075404.1	82.20%	2 × 10^−54^
20	*Uncharacterized LOC8283164*	XM_002524251.3	80.15%	7 × 10^−45^
21	*Protein argonaute 4-like*	XM_028251957.1	86.89%	0.0
25	*Serine/threonine-protein kinase AFC2*	XM_027915241.1	79.02%	5 × 10^−74^
26	*protodermal factor 1-like*	XM_028230087.1	77.40	1 × 10^−47^

## Data Availability

Not applicable.

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
