# Peer review of "A Study of the Molecular Regulatory Network of VcTCP18 during Blueberry Bud Dormancy"

_plants, 2023, doi:10.3390/plants12142595_

Round 1

Reviewer 1 Report

Dear Authors,

It was with pleasure that I read your manuscript entitled: ‘Study of molecular regulatory network of VcTCP18 in Blueberry bud dormancy’.

The authors show a nice analysis of TCP18 function in three different blueberry cultivars, with varying degrees of axillary bud growth and branching phenotypes. To try and explain the different branching and cold requirement phenotypes the authors find TCP18 expression patterns that explain the phenotypes. Following, the authors analyse the promoter region, trying to identify regulatory elements that could explain the TCP18 expression pattern and response to cold.

There are a few points I’d like to raise regarding the experimental setup. When it comes to the results, conclusions, and discussion I have only a few minor comments that I shall outline while referring to the sentences.

I would like the authors to explain their choice of promoter region, and the subsequent fragments of this promoter the authors use in the Y1H and LUC-assay. 

To start with, there’s no justification for the chosen 2860bp fragment. Although, by a dual luciferase assay, the authors show promoter activity, I would prefer the authors to elaborate on their choice of region. When trying to elucidate the activity of sub-regions of the promoter, the authors write ‘different promoter lengths were deleted based on the distribution of key cis-regulatory elements’ with no elaboration on which cis-regulatory element the authors refer to. Please provide more detail.

I would strongly suggest a complementation essay; since the expression pattern of VcTCP18 is nearly identical to Arabidopsis TCP18/BRC1 I would think that providing an Arabidopsis mutant complemented with a VcTCP18-promoter driven BRC1 would be the ultimate proof that the choice of promoter is justified. 

In a follow-up to the Luciferase essay with three promoter regions, the authors perform yeast-one-hybrid with the VcTCP18 promoter to try and identify genes binding the promoter and possibly regulating TCP18. I don’t understand the choice of promoter regions, here the authors skip the smaller region (-1595 to -1). Indeed, this region seems not responsible for the difference in response to temperature, as the authors show in figure 4. However, the region is active, as shown by the luciferase assay, and the Y1H is performed on a cDNA library that has no relation to different temperatures. I therefore believe that the authors might miss regulatory elements in their Y1H.

Line 100 and 102. Please explain the abbreviations NHB and SHB the first time they are mentioned here.

Line 105 and the description of figure 1. Please explain which tissue was used, axillary buds, buds, whole plants?

Line 167. ‘Be owing to many’ confused me, what are the authors trying to say? Many low temperature response elements were found in the promoter? If so, which ones, and how are they distributed?  

Line 180-181. For clarity I would suggest to explain the different promoter elements used. If I get it correctly T1 and T2 are the -2860 to -2310 and -2310 to -1595 respectively.

Line 211. Almost all of the…?

Line 251-253. I think ABA should be mentioned in this sentence as well, as an inhibitor of branching/bud break?

Line 271-272. If the cultivar ‘Emerald’ exhibits no difference in VcTCP18, how do the authors explain a higher lateral budbreak rate? 

Line 314. p2VcBRC1, is probably supposed to be p2VcTCP18?

All in all, the story is rather solid; the authors find confirmation of the role of the master regulator of branching, TCP18, in different blueberry cultivars, under different treatments. I think the authors could focus a little more on the Y1H results, as this seems to be a novelty as compared to the expression patterns, and branching phenotypes that are all well described in different species. 

In conclusion, I would suggest that when the revisions are met, the manuscript is fit for publication in ‘plants’.

Best,

see the minor points in the 'suggestions for authors'.

Author Response

please see in attachment

Reviewer 2 Report

The manuscript by Li et al. presents interesting results correlating the expression level of blueberries (Vaccinium corymbosum) TCP18 gene with the chilling requirements and bud dormancy release. Using the dual luciferase assay, the authors verified that low temperature inhibits the expression driven by the VcTCP18 promoter. Additionally, they screened a blueberry flower bud cDNA library in the yeast one-hybrid system, to identify putative transcription factors binding to the TCP18 promoter and regulating its expression. The authors also studied the expression profiles of VcTCP18 and its potential upstream regulatory during flower bud dormancy release. Considering the results obtained and the data available at the literature, the authors propose a molecular regulatory network of VcTCP18 in blueberry bud dormancy.

Points to be considered:
1) The English should be revised, since there are incomplete sentences, with words missing. E.g. Line 51: “The similar phenomenon also observed in sweet cherry…”
2) Please include the word “dormancy” in the expression “Bud dormancy break” in the different places throughout the text.
3) Line 69: Please, define SD.
4) Please, correct the year at line 91
5) Line 94: Please rewrite and better explain the text “…, its axillary buds break vigorous, …”
6) Please include a model explaining times and dormancy in blueberry in Figure 1, since (2d, Nov 17, Nov 19, Dec 8 and Dec 29; 6h, 12h, 18h and 24h from Figure 6) do not mean much for researchers not familiar with blueberries.
7) On line 143, Table S3 is mentioned. However, Tables S1 and S2 were not presented before.
8) Line 147: Please, rewrite the text “… a length of 2860bp promoter…” Suggestion: “… the 2860 bp sequence above the ATG (here defined as the promoter)…”
9) Please review the separation of syllables/words in the lines 152 and 153.
10) Line 162: Please, revise the use of VcBRC1 instead of VcTCP18 throughout the text.
11) Please, correct the word vector on line 166.
12) Please, revise the 2 sentences on lines 167-169.
13) Item 2.3, Screening of a yeast one-hybrid library: I would consider removing the clones encoding class III chitinase, glyceraldehyde-3-phosphate dehydrogenase GAPCP1, and selenocysteine methyltransferase from the results. These proteins are not transcription factors, are not discussed on the manuscript and could be considered as “sticky” proteins frequently found as false-positives in Y2H/Y1H screenings.
14) Line 199: The same type of verification of positive clones was described with different names: “Rotation verification”; “gyration verification”; and “reversal validation” – Figure 5. Please, standardize the term throughout the manuscript.
15) Item 2.5, Expression analysis of screened genes: Please, correlate the descriptions on lines 217-225 with the hours that appear in the expression Figure 6, as not all researchers know how processes take place in blueberry.
16) Please, improve the descriptions at the Figure legends 5, 6 and 7.
17) Image on Figure 5: Assays without and with aureobasidin could not have been performed on the same plate, could they? Please remake the figure, including proper identification of the tested clones. Not all are TFs!
18) Lines 291-295: Please, define/explain DAM.
19) Please, review and improve the wording of the sentences in the lines 295-300, to make the ideas clearer.
20) Lines 319-320: It is written: “In this study, a total of 21 genes were screened using yeast one-hybrid screening, …” However, I believe that a very large number of clones have been screened and 21 clones have been revealed as positive, is not it? Please, rewrite for clarification.
21) Lines 319-326: The T2-17 AP2-like ERF from Figure 6 corresponds to ANT (AINTEGUMENTA)? Please, standardize the nomenclature and present an adequate description of the genes under study, especially in Figures 5 and 6.
22) Please, revise the sentence on lines 334-335 for clarification.
23) Lines 364-374: Please, clarify the proposal. Are the authors proposing that dehydrin binds to the VcTCP18 promoter or that dehydrin and TCP18 are co-expressed? In my opinion, the proposal presented is very speculative and there is not much data to support it.
24) Lines 391-397: Please, revise and rewrite the sentences.
25) Item 4.2, qRT-PCR: In my opinion, authors should report the amount of initial RNA used for cDNA synthesis.
26) Lines 437-438: Please, revise and rewrite the sentence.
27) Lines 444-445: Please, revise and rewrite the sentence. Monoclonal of the positive bait…?

no

Author Response

please see in attachments

Reviewer 3 Report

The study explores the expression characteristics and molecular regulatory mechanism of BRANCHED1 (BRC1) in blueberry bud dormancy and sheds light on the correlation between chilling requirements (CR) and tree shape. Key takeaways from this research are:

1. The high CR cultivar 'Chandler' had the strongest apical dominance among the three cultivars, and the expression of the VcTCP18, which is homologous to BRC1, was the highest in both treatments.

2. The 'Emerald' cultivar, with a low CR, demonstrated the opposite trend, suggesting that VcTCP18 plays a negative regulatory role in bud break.

3. Yeast 1hybrid (Y1H) assays screened out 21 upstream regulatory genes, including eight transcription factors, that can positively or negatively regulate the expression of VcTCP18 through transcriptome expression profile.

The text of the paper is factual concrete, realistic, and understandable. But, there are important flaws in the manuscript listed below:

1-    The manuscript has major grammar and punctuation problems such as in line 16 however should be replaced by However; in line 413 “Total DNA was extracted from blueberry flower buds using the CTAB method [45], 413 DNA was stored at -20 °C”; should replace with Total DNA was extracted from blueberry flower buds using the CTAB method [45] and was stored at -20 °C”.

2-    It also needs to be checked for the English language by a native speaker.

3-    In the plant material, more explanation about three blueberry cultivars- 'O'Neal' and 'Emerald' 'Chandler’ and references for their chilling requirements are needed.

4-    In Figures  1, 2 and 4, the extra explanation about analysis methods could be removed and this explanation can be added to the material and methods.  

5-     The study does not provide a comprehensive understanding of the regulatory network of BRC1 in bud dormancy and further research is needed to fully understand its function.

In light of the above consideration, I suggest a major revision of the manuscript.

The manuscript has major grammar and punctuation problems and needs to be checked for the English language by a native speaker.

Author Response

please see in attachment

Round 2

Reviewer 3 Report

The authors have considered all points in the revised manuscript and therefore, it can be accepted for publication.